# Twinning in Zr-Based Metal-Organic Framework Crystals

**Sigurd Øien-Ødegaard ***  **and Karl Petter Lillerud**

Centre for Materials Science and Nanotechnology, University of Oslo, P.O. box 1126 Blindern, 0318 Oslo, Norway
* Correspondence: sigurdoi@kjemi.uio.no

**Abstract:** Ab initio structure determination of new metal-organic framework (MOF) compounds is generally done by single crystal X-ray diffraction, but this technique can yield incorrect crystal structures if crystal twinning is overlooked. Herein, the crystal structures of three Zirconium-based MOFs, that are especially prone to twinning, have been determined from twinned crystals. These twin laws (and others) could potentially occur in many MOFs or related network structures, and the methods and tools described herein to detect and treat twinning could be useful to resolve the structures of affected crystals. Our results highlight the prevalence (and sometimes inevitability) of twinning in certain Zr-MOFs. Of special importance are the works of Howard Flack which, in addition to fundamental advances in crystallography, provide accessible tools for inexperienced crystallographers to take twinning into account in structure elucidation.

**Keywords:** MOFs; crystallography; twinning

## 1. Introduction

Metal-organic frameworks (MOFs) are porous solids consisting of inorganic nodes linked by organic multidentate ligands (e.g., through strongly coordinating groups such as carboxylates or Lewis bases) to form extended networks [1]. The combination of diverse coordination chemistry and the structural richness of organic ligands enable a vast number of possible MOF structures and network topologies [2]. Due to their remarkable ability to harbor a large range of chemically functional groups in pores that are accessible to guest species, MOFs are studied mainly (albeit not exclusively) for their properties as adsorbents [3,4] and catalysts [5]. In particular, MOFs based on zirconium oxide clusters are frequently reported due to their relatively high stability and structural diversity [6]. These MOFs are normally obtained as single crystals by adding growth modulators (monocarboxylic acids) to the synthesis liquor [7].

The principal method of MOF structure determination is X-ray diffraction methods, and single crystal X-ray diffraction (SC-XRD) in particular [8]. Despite being the most powerful technique to solve crystal structures, a successful SC-XRD experiment normally depends on a relatively large single crystal of high quality.

The phenomenon in which a crystal consists of two or more separate domains, and these domains are related by a symmetry operation that is not present in the space group of the crystal, is known as twinning [9]. Twinned crystals represent a commonly encountered problem in crystallography and will often prevent a successful structure determination. The observed reflections of all the present domains can be interpreted as originating from the same crystal instead of separate entities, thus obscuring the true symmetry (and thus the structure) of the crystal. Crystalline MOFs and related network structures are prone to interpenetration (intergrown separate networks that may occur if the void fraction of the structure is large) and twinning, because twin domain interfaces are enabled by the flexibility of the MOF's building units (i.e., geometrical flexibility of (1) the linkers' points of connectivity due to

free rotation around C-C bonds and (2) the possible existence of metal oxoclusters with a different arrangement of coordination sites in twin boundaries).

In our ongoing crystallographic characterization of Zr-based MOFs, several cases of crystal twinning or related problems were discovered, which can be categorized into three groups:

(a) A crystal consisting of two or more domains related by seemingly arbitrary rotation matrices (although not technically twinning), is frequently encountered when investigating MOF crystals. It occurs when two randomly oriented crystals in close proximity grow into each other and forms an interface, and is frequently observed in static syntheses where crystal growth mainly occurs on the bottom of the synthesis vessel. In these cases, automatic indexing fails to provide a meaningful unit cell, but the relationship can usually be determined by manual inspection and sorting of the reflections in reciprocal space.

(b) In certain cases, automatic indexing found a hexagonal supercell due to partial overlap between the reflections from the twin domains. The twin law was found to be the so-called "spinel law", $2_{[111]}$ (where the two twin domains are related by a two-fold rotation about the body diagonal of a cubic unit cell), which is a case of twinning by reticular merohedry [10]. This twinning mode was observed in UiO-67, UiO-67-Me$_2$ (1, discussed herein) and Zr-stilbene dicarboxylate. In all of these cases, the crystals displayed a specific morphology, resembling intergrown octahedra with two shared (1 1 1) faces (Figure 1).

(c) In certain cases, twinning by syngonic merohedry was observed as a consequence of intrinsic features of the MOF. The examples presented herein are obtained from MOFs featuring partial lattice interpenetration (3) and a phase transitions from dynamic to static orientation disorder upon cooling of the sample (2).

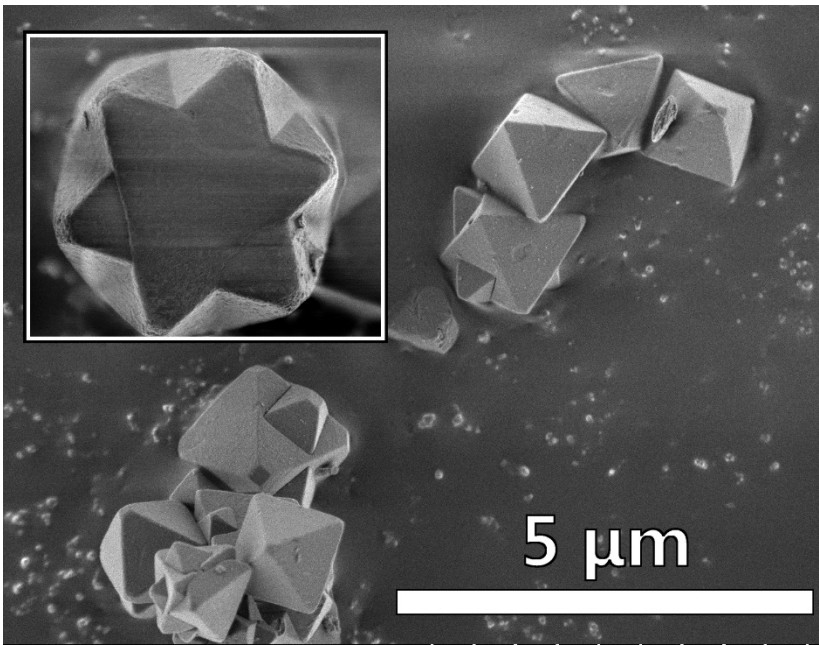

**Figure 1.** Crystals of UiO-67-Me2, showing the presence of twinning by the spinel law.

The following sections describe how these twinning issues has been resolved for specific Zr-MOF samples, and how a correct assessment of twin laws has revealed structural features with significant implication for the MOFs' properties. The strategies described herein to detect and resolve twinning will presumably be useful to researchers encountering the issue while working with MOFs and other crystalline network solids.

## 2. Materials and Methods

### 2.1. General Materials and Synthesis Methods

All chemicals, unless otherwise stated, were obtained reagent grade from Sigma-Aldrich and used without further purification. The linkers for 1 and 2 (3,3′-dimethyl-4,4′-biphenyldicarboxylic acid (Me$_2$-H$_2$bpdc) and 2′,3′′-dimethyl-[1,1′:4′,1′′:4′′,1′′′-quaterphenyl]-4,4′′′-dicarboxylic acid (Me$_2$-H$_2$qpdc)) were prepared using previously published protocols [11]. Crystallization was performed under static conditions using conical glass flasks that were treated with NaOH$_{(aq)}$ 30% wt. overnight then thoroughly rinsed and dried before the reaction. This treatment seems to decrease the MOF's tendency to nucleate rapidly on the glass surface. During crystallization the flasks were capped with loose lids to prevent pressure build-up and accumulation of decomposition products of DMF (mainly formic acid and dimethylamine) and HCl.

Synthesis of UiO-67-Me$_2$ (1): To 13 mL of dimethylformamide (DMF), 56 µL (3.1 mmol) H$_2$O, 241 mg (1.03 mmol) ZrCl$_4$, and 3.79 g (31.0 mmol) benzoic acid was added and stirred until a clear solution was obtained, and subsequently heated to 120 °C. 280 mg (1.03 mmol) Me$_2$-H$_2$bpdc was then added, a clear solution was quickly obtained, transferred to a clean conical flask and kept at 120 °C for 72 h. The solid crystalline product was isolated and washed, once briefly with DMF at 100 °C, once with dry DMF at room temperature and three times with dry 2-propanol. The MOF crystals were kept in dry 2-propanol prior to activation and XRD measurement.

Synthesis of Zr-stilbene dicarboxylate (2): To 50 mL of DMF, 70 µL (3.9 mmol) H$_2$O, 301 mg (1.29 mmol) ZrCl$_4$ and 4.73 g (38.8 mmol) benzoic acid was added and stirred until a clear solution was obtained, and subsequently heated to 120 °C. 346 mg (1.29 mmol) trans-stilbene-4,4′-carboxylic acid was then added, a clear solution was obtained and the solution was transferred to a clean conical flask and kept at 120 °C for 72 h. Large octahedral crystals were isolated and washed. First briefly with DMF at 100 °C, then dry DMF at RT, then three times with tetrahydrofuran (THF), and lastly three times with n-hexane. The crystals were kept in dry n-hexane prior to measurement.

Synthesis of UiO-69-Me$_2$ (3): To 20 mL of DMF, 19 µL (1.03 mmol) H$_2$O, 120 mg (0.52 mmol) ZrCl$_4$ and 1.89 g (15.5 mmol) benzoic acid was added and stirred until a clear solution was obtained, and subsequently heated to 120 °C. 218 mg (0.52 mmol) Me$_2$-H$_2$qpdc was then added, a clear solution was obtained, transferred to a clean conical flask and kept at 120 °C for 72 h. The solid crystalline product was isolated and washed.

### 2.2. X-ray Crystallography

The crystals of 1 and 3 were dried in air at 200 °C for 2 h prior to measurement to ensure that the pores were free of physiosorbed solvent and water. The crystals were mounted on MiTeGen polymer loops using a minimum amount of Paratone oil. Data collection for 1–3 was performed on a Bruker D8 Venture diffractometer equipped with a Photon 100 CMOS detector using Mo K$\alpha$ radiation ($\lambda$ = 0.71073 Å) at 100 K. 3 was also measured at beamline ID11 at the ESRF synchrotron (Grenoble, France), equipped with a Frelon2 detector, using a wavelength of $\lambda$ = 0.31120 Å.

Due to the poor chemical stability of 2, it was kept in dry hexane after synthesis. Before measurement, several crystals were placed on a glass slide and the hexane was allowed to evaporate. The crystals were then mounted on MiTeGen polymer loops using a minimum amount of Paratone oil. 2 was also measured at beamline I911-3 at the MAX2 synchrotron (Lund, Sweden) [12].

All frames were integrated, and the reflection intensities were scaled and evaluated using the APEX3 suite from Bruker AXS, consisting of SAINT, SADABS and XPREP [13]. The structures were solved with XT [14] and refined with XL [15], using Olex2 as graphical user interface [16]. Space group determination of twinned MOFs was done using comprehensive tables by Howard Flack [17]. A summary of the crystal data, data collection and structure refinement details can be found in Table 1, and full structural information can be found in the Supplementary Materials.

**Table 1.** Summary of crystallographic data and refinement indicators for reported MOFs 1–3.

| Crystal Data | UiO-67-Me$_2$ (1) | Zr-Stilbene dc (2) | UiO-69-Me$_2$ (3) |
|---|---|---|---|
| Chemical formula | $C_{384}O_{128}Zr_{24}\cdot32(O)$ | $C_{342.25}H_{211.09}O_{128}Zr_{24}$ | $C_{648}H_{408}O_{128}Zr_{24}\cdot$ $0.66(C_{648}H_{408}O_{128}Zr_{24})\cdot$ $16(O)$ |
| M$_r$ | 2340.28 | 8584.82 | 20,698.65 |
| Crystal system, space group | Cubic, Fm$\overline{3}$m | Cubic, Pn$\overline{3}$ | Cubic, F$\overline{4}$3m |
| Temperature (K) | 100 | 100 | 100 |
| a (Å) | 26.8903 (12) | 30.0322 (6) | 38.995 (2) |
| V (Å$^3$) | 19,444 (3) | 27,087.0 (16) | 59,298 (10) |
| Radiation type | Mo K$\alpha$ | Synchrotron, $\lambda$ = 0.760 Å | Synchrotron, $\lambda$ = 0.3112 Å |
| $\mu$ (mm$^{-1}$) | 0.35 | 0.25 | 0.17 |
| Crystal size (mm) | $0.06 \times 0.06 \times 0.02$ | $0.2 \times 0.2 \times 0.2$ | $0.14 \times 0.14 \times 0.14$ |
| **Data collection** | | | |
| Diffractometer | Bruker D8 Venture, CMOS detector | MD2 microdiffractometer with MK3 mini-kappa | ESRF ID11 |
| Absorption correction | Multi-scan | Multi-scan | Multi-scan |
| No. of measured, independent and observed [I > 2σ(I)] reflections | 4388, 4388, 4155 | 199,694, 11,196, 10,761 | 180,671, 19,944, 17,103 |
| R$_{int}$ | 0.036 | 0.04 | 0.033 |
| $(\sin\theta/\lambda)_{max}$ (Å$^{-1}$) | 0.649 | 0.623 | 0.961 |
| **Refinement** | | | |
| R[F$^2$ > 2σ(F$^2$)], wR(F$^2$), S | 0.048, 0.174, 1.13 | 0.029, 0.090, 1.10 | 0.047, 0.154, 1.08 |
| No. of reflections | 4388 | 11,196 | 19,944 |
| No. of parameters | 59 | 216 | 226 |
| No. of restraints | 0 | 18 | 304 |
| $\Delta\rho_{max}$, $\Delta\rho_{min}$ (e Å$^{-3}$) | 1.03, −1.05 | 0.91, −0.46 | 2.17, −2.60 |

## 3. Results and Discussion

### 3.1. UiO-67-Me$_2$

UiO-67-Me$_2$, consisting of the Zr$_6$ oxocluster and the Me$_2$-bpdc linker, is isostructural to UiO-67, with nearly identical lattice parameters and PXRD pattern. The MOF readily forms large single crystals during synthesis, by the addition of 30 molar equivalents of benzoic acid (in respect to Zr) as modulator. This MOF has been found to have slightly better stability to water, and higher affinity to methane than UiO-67, presumably due to the steric shielding of cluster-adjacent hydrophilic sites by the methyl groups [18].

A significant number of the crystals that were screened had visibly distorted morphology (Figure 1). The crystals were seemingly intergrown, as if one crystal was sprouting out of the facets of its parent. In all of these crystals, the initial indexing resulted in a larger unit cell than expected, a primitive hexagonal cell, closely related to the reduced cell of UiO-67. (Reduced unit cell of UiO-67: a = b = c = 19.0 Å, α = β = γ = 60.0°. Hexagonal cell: a = b = 19.0 Å, c = 47.0 Å, a = b = 90°, c = 120°).

Upon inspection of the reciprocal lattice, it was apparent that the crystals were twinned by reticular merohedry, the twin law being the so-called "spinel law", 2$_{[111]}$. This twin law translates to a two-fold rotation about the [1 1 1] axis, coinciding with a three-fold rotation-reflection axis (S$_6$ or $\overline{3}$) in the ideal structure. This is a common twinning mode in cubic close packed crystals, as it is associated with stacking error of the close packed layers.

The twin law is easily visualized by reciprocal lattice displays of the reflections of the hk0 layer (Figure 2). The cause for twinning or the exact structure at the interface of this MOF remains unknown, but there are other Zr-based oxoclusters that facilitate such a boundary [19].

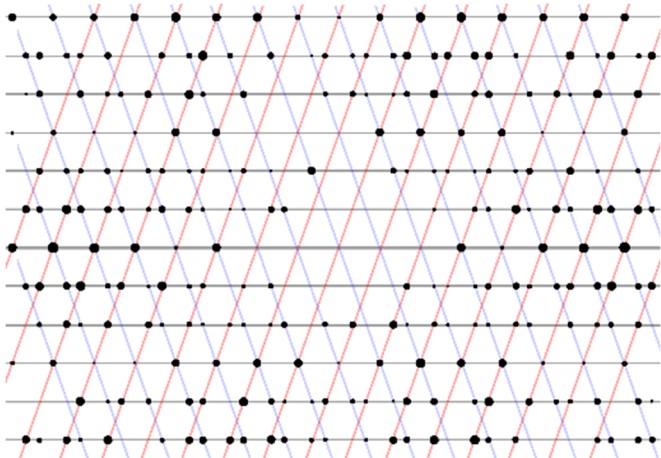

**Figure 2.** Schematic display of the hk0 layer of the diffraction of 1, twinned by $2_{[111]}$. The lattices of the twin domains are shown in red and blue, respectively.

### 3.2. Zr-stilbene Dicarboxylate

Zr-stilbene dicarboxylate (2) has previously been reported in its disordered form (space group Fm-3 m), in which the crystal symmetry is the same as for the UiO MOFs [20,21]. These MOFs consist of the same $Zr_6$ oxoclusters in an fcc arrangement, connected by linear ditopic linkers. However, the stilbene linker (being non-linear) is not compatible with this symmetry, implying that it is randomly disordered over two rotational conformers. The disorder could be dynamic, in which the linkers would rotate freely and continuously throughout the sample because of the low energy barrier of rotation. In this case, one would expect a threshold temperature at which the rotation is no longer feasible, and where the individual linkers adopt a static conformation. Such a static phase could exist with linkers assuming random conformers (preserving the overall $Fm\overline{3}$ m disordered symmetry) or ordered conformations resulting in a symmetry change.

Repeated measurements at room temperature and after flash cooling samples to 100 K showed the randomly oriented linker structure ($Fm\overline{3}$ m) previously reported [21]. To investigate whether a low-symmetry phase could be obtained, individual scans along [1 0 0] were acquired while slowly cooling the crystal by 5 K/min. The appearance of new diffraction peaks (forbidden in face centered cubic crystals) occurred around 160 K (Figure 3).

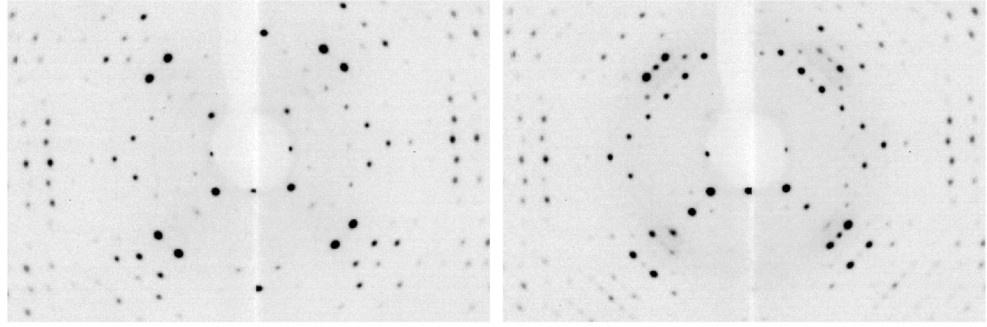

**Figure 3.** Oriented diffraction images of **2** along the [1 0 0] direction acquired at room temperature (**left**) and at 100 K (**right**) after slow cooling of the crystal (by 5 K/min).

Solving the structure using the data from the slowly cooled crystal revealed a structure in the primitive cubic space group Pn$\bar{3}$ (shown in Figure 4a–c). In this structure, the conformation of the linkers are unambiguous; the space group does not include mirror planes intersecting the linkers as in Fm$\bar{3}$ m. When the crystal is brought back to room temperature, the face centered phase is observed again (shown in Figure 4d–f). This reversible phase transition can be used to study the molecular mechanics of these nonlinear linkers, which have also been shown to facilitate linker exchange and lattice expansion in these MOFs [22].

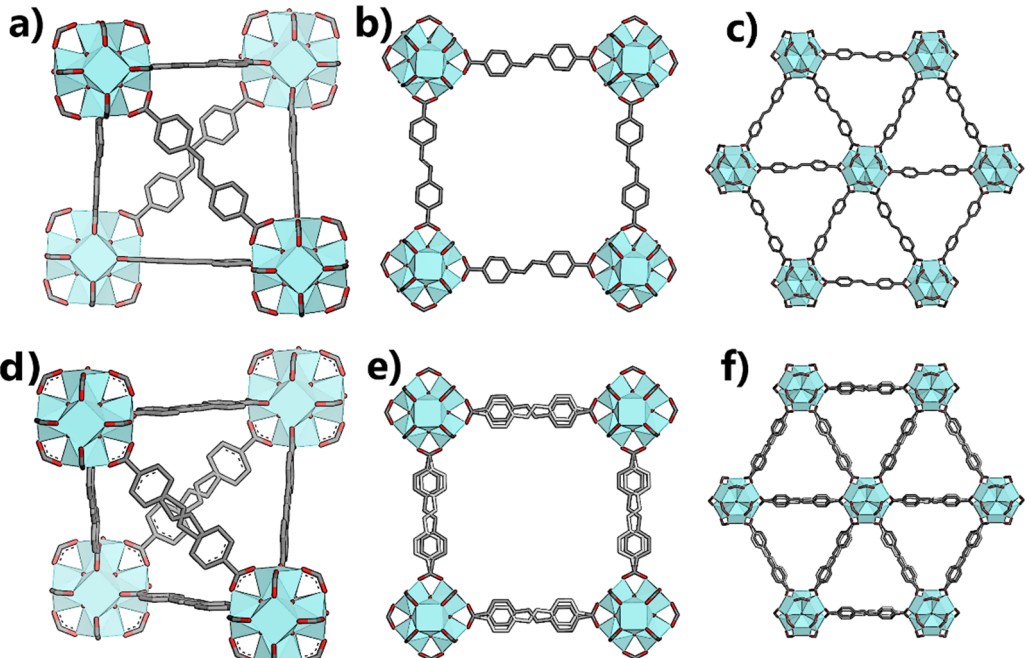

**Figure 4.** Partial crystal structures of **2** in Pn$\bar{3}$ (**a**–**c**) and Fm$\bar{3}$ m (**d**–**f**) showing the ordered and disordered conformations of the linker.

Inspecting the initial solution of the Pn$\bar{3}$ structure, it displayed several warning signs: Very high *R*-values and presence of significant noise in the Fourier difference maps, large negative peaks in particular. Looking at the crystal structure along [1 0 0], one could envision two different orientations of the linker "kink" being present in the crystal (Figure 5). In fact, are separate domains of the crystal with opposite oriented linkers related to each other by a two-fold rotation about the [1 1 0] face diagonal. Thus, the twin law to implement in the crystal structure refinement is the matrix (0 −1 0 −1 0 0 0 0 −1). Applying this twin law to the refinement significantly improves the fit, and the twin fractions refine freely to 0.5.

### 3.3. Interpenetrated UiO-69-Me$_2$

UiO-69-Me$_2$ (3) was synthesized as single crystals using the linker 2′,3′′-dimethyl-[1,1′:4′,1′′:4′′, 1′′′-quaterphenyl]-4,4′′′-dicarboxylic acid (Me$_2$-H$_2$qpdc) [11]. Using 30 equivalents of benzoic acid in respect to ZrCl$_4$ as modulator, single crystals of up to 1 mm could be obtained. The large octahedral single crystals featured a distinct opaque pattern originating from the center of each crystal and out to each face (Figure 6).

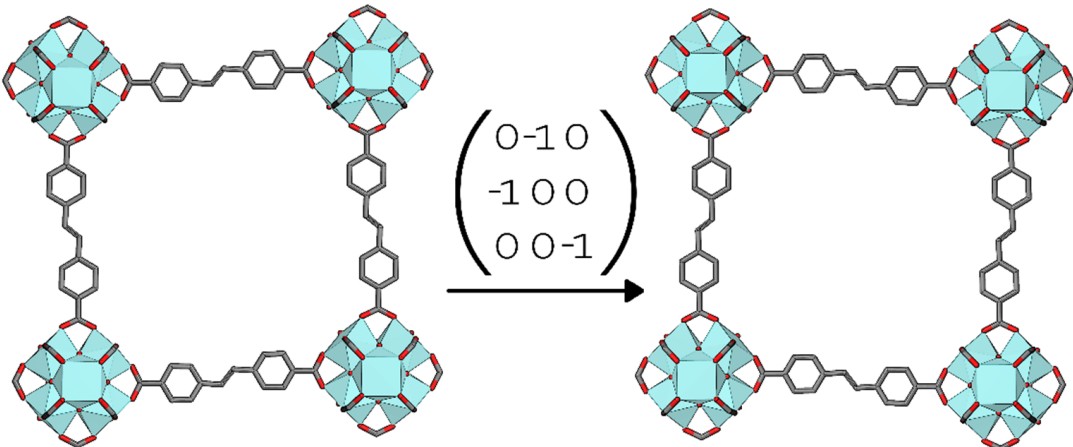

**Figure 5.** The twin law of 2 in the Pn$\bar{3}$ space group.

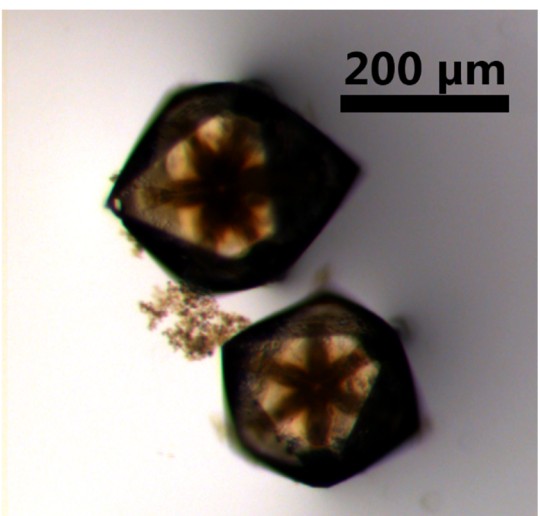

**Figure 6.** Single crystals of 3, featuring an opaque pattern from the center and out towards each face.

High quality SC-XRD data was acquired using synchrotron radiation, and the subsequent structure solution revealed a structure composed of doubly interpenetrated fcu networks analogous to the PIZOF materials [23]. Based on analysis of reflection intensities, the structure appears to have the space group Fd$\bar{3}$ m, as is observed for the other family of two-fold interpenetrated Zr-MOFs [24]. The structure could indeed be solved in this space group, but the refinement accuracy came out very poor ($R_1 > 13.5\%$).

When a structure appears centrosymmetric it could in reality be a twin of two equally large non-centrosymmetric domains [9,25]. Consequently, the structure was solved in the lower-symmetry space group F$\bar{4}$3m. The difference between Fd$\bar{3}$m and F$\bar{4}$3m is one symmetry element, a two-fold rotation relating the closest neighboring clusters from the two MOF lattices (intersecting 1/8 1/8 1/8). When the symmetry element is present, the two lattices must be identical for the symmetry to be true. However, this MOF feature only partial interpenetration, so the secondary lattice has a lower occupancy coefficient than the main lattice (in this case 0.66). When this coefficient is not equal, the real symmetry must be the lower F$\bar{4}$3m.

The reduction in symmetry reveals a curious detail: Neighboring clusters from the separate lattices have a clear tendency to orient opposite $\mu_3$-O/OH functionality towards each other. The fully occupied lattice tend to orient the $\mu_3$-O towards the partially occupied neighboring cluster's OH. This preference would violate the two-fold rotational symmetry of Fd$\bar{3}$m. There are two possible settings of the secondary lattice within the main, which requires a twin law (e.g., twinning by inversion)

to resolve (Figure 7). This twinning feature contributes to the high apparent symmetry of the data, leading to the initial erroneous space group assignment.

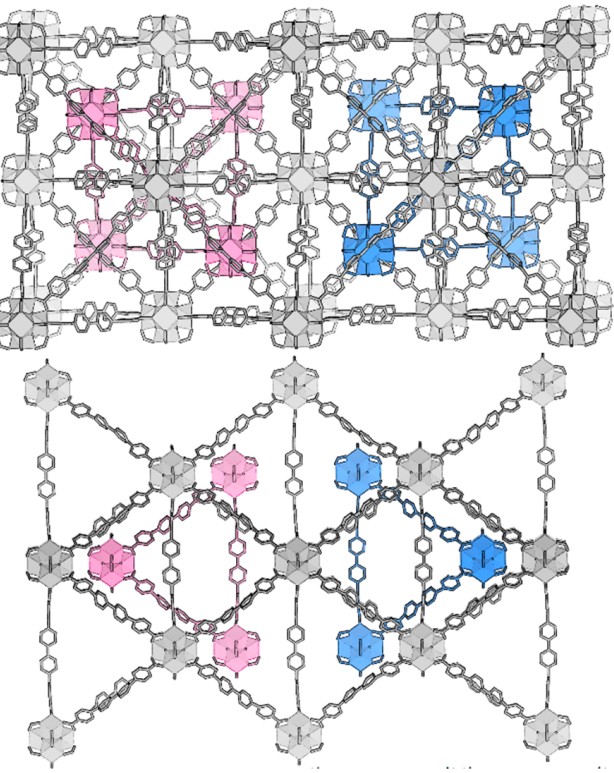

**Figure 7.** The twinning in UiO-69-Me$_2$ (3) can be visualized by displaying the two possible settings of the interpenetrating lattice.

## 4. Conclusions

The results reported herein clearly show that twinning should always be considered in the single-crystal structure determination of MOFs and other network structures, in particular in cases where unexpected features arise during structure solution and/or refinement. Details such as partial interpenetration or structural heterogeneity within the sample may easily be overlooked if twinning is not considered. It is likely that several of the many reported MOF structures that display poor refinement values or chemically unlikely features are in fact non-treated twins.

**Supplementary Materials:** The following are available online at http://www.mdpi.com/2624-8549/2/3/50/s1, Crystal structure data of compounds **1**–**3**. The crystallographic data for this paper (CCDC) can also be obtained free of charge via www.ccdc.cam.ac.uk/data_request/cif, by emailing data_request@ccdc.cam.ac.uk, or by contacting The Cambridge Crystallographic Data Centre, 12 Union Road, Cambridge CB2 1EZ, UK; Fax: +44-1223-336033.

**Author Contributions:** Conceptualization, S.Ø.-Ø.; methodology, S.Ø.-Ø.; validation, S.Ø.-Ø.; formal analysis, S.Ø.-Ø.; investigation, S.Ø.-Ø.; resources, S.Ø.-Ø.; data curation, S.Ø.-Ø.; writing—original draft preparation, S.Ø.-Ø.; writing—review and editing, S.Ø.-Ø.; visualization, S.Ø.-Ø.; supervision, K.P.L.; project administration, K.P.L.; funding acquisition, K.P.L. All authors have read and agreed to the published version of the manuscript.

**Funding:** This research was funded by the European Union's Horizon 2020 research and innovation programme under grant agreement number 685727 (ProDIA).

**Acknowledgments:** Knut T. Hylland is acknowledged for providing the organic linkers for the synthesis of compounds **1** and **3**.

**Conflicts of Interest:** The authors declare no conflict of interest.

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

**Sample Availability:** Not available.

