# Peer review of "Twinning in Zr-Based Metal-Organic Framework Crystals"

_chemistry, doi:10.3390/chemistry2030050_

Round 1

Reviewer 1 Report

The authors have determined the crystal structures of three zirconium-based MOFs that are prone to twins using twin crystals. The authors suggested that detecting and treating twins using the methods and tools described in the manuscript could be useful in resolving the structure of the affected crystals. Although the results of the study on the crystal structure showed high reliability, the information in the manuscript was limited to the reader's understanding of the author's proposal.

In the introduction, the author does not have a description of how what he wants to do in this study improves or solves problems in the existing study, and the specific details he wants to do in this study are not specified. Accordingly, the reviewer only estimates the purpose of the content of the study based on the abstract, which is limited in conveying the meaning of the results of this study to readers. Therefore, in this introduction, a more specific background explanation of what the author is trying to solve through this study is necessary.

 In addition, I have read many times to understand the scientific implications of manuscripts and how they will contribute to future research, but it was difficult to clearly grasp the proposals the author emphasized.

 I trust the author's findings, but it's difficult to determine if this is a good fit for a chemistry journal. Accordingly, I would like to follow the opinions of other reviewers in the decision to publish the paper.

The information below may be useful for future revisions or other journal entries.

1) References have been excluded from many of the sentences in the Introduction. Accordingly, it is essential to add a reference for the reader in the sentence below.

-Line 22: Metal-organic frameworks (MOFs) are porous solids consisting of inorganic nodes linked by organic multidentate ligands (e.g. through strongly coordinating groups such as carboxylates or Lewis bases) to form extended networks.

-Line 24: The combination of diverse coordination chemistry and the structural richness of organic ligands enable a vast number of possible MOF structures and network topologies.

-Line 32: The principal method of MOF structure determination is X-ray diffraction methods, and single crystal X-ray diffraction (SC-XRD) in particular.

-Line 33: Despite being the most powerful technique to solve crystal structures, a successful SC-XRD experiment normally depends on a relatively large single crystal of high quality.

2) I suggest a consistent expression in the two sentences below. This will be useful to increase the readability of the reader.

Line 104: …beamline ID11 at the ESRF synchrotron 104 (Grenoble, France),

Line 109: … beamline I911-3 at the MAX2 synchrotron in Lund, Sweden.

3) In Figure 4, the notation (d) overlaps with the figure 4d.

4) Line 187: “… acquired at beamline ID-11 at the European synchrotron (ESRF)” This sentence has already been mentioned in the materials and methods section.

5) The reference must appear at the end of the sentence and before the period.

6) Authors should clearly indicate the references below.
Reference 4: The author should indicate the volume and page number. The word "doi" was duplicated.

Reference 6: “doi” was duplicated.

Reference 11-14: “doi” was duplicated.

Reference 19: The author should remove “http://doi.org/”

Reference 22: below sentence should be deleted . “http://www.nature.com/nchem/journal/v8/n3/abs/nchem.2430.html#supplementary-information.”

7) In order to further emphasize the author's scientific results, it will be important to cover more details about the meaning of this study and how it can be applied to crystal twin in the future in discussion and conclusion.

Author Response

We are pleased to submit the revised manuscript entitled “Twinning in Zr-based Metal-organic Framework Crystals”. We would like to thank the reviewers, who have provided insightful and constructive criticism which have improved the clarity and quality of the manuscript. Below is a point-by-point summary of the reviewer’s remarks and our replies.

Reviewer 1:

The authors have determined the crystal structures of three zirconium-based MOFs that are prone to twins using twin crystals. The authors suggested that detecting and treating twins using the methods and tools described in the manuscript could be useful in resolving the structure of the affected crystals. Although the results of the study on the crystal structure showed high reliability, the information in the manuscript was limited to the reader's understanding of the author's proposal.

In the introduction, the author does not have a description of how what he wants to do in this study improves or solves problems in the existing study, and the specific details he wants to do in this study are not specified. Accordingly, the reviewer only estimates the purpose of the content of the study based on the abstract, which is limited in conveying the meaning of the results of this study to readers. Therefore, in this introduction, a more specific background explanation of what the author is trying to solve through this study is necessary.

In addition, I have read many times to understand the scientific implications of manuscripts and how they will contribute to future research, but it was difficult to clearly grasp the proposals the author emphasized.

I trust the author's findings, but it's difficult to determine if this is a good fit for a chemistry journal. Accordingly, I would like to follow the opinions of other reviewers in the decision to publish the paper.

Reply:  From our viewpoint, this is more of a thematic report than a pre-designed scientific study, simply because we could not have designed MOF crystals for twinning. The reported structures were encountered during studies of MOF compounds for other purposes. However, we think that the results and findings we report still have a place in the MOF and crystallographic literature, as it provides new insight in the structural complexity of previously reported MOFs. We also think our strategies to detect and resolve twinning can be applied to MOFs and crystalline network structures in general.

There are several examples in the literature of MOF structures that are probably wrong, because twinning has been overlooked in the data reduction. However, we cannot be sure of what mistakes have been made without having seen the raw data. For this reason, we decided not to name any specific publications.

However, we have appended a paragraph to the introduction to clarify what the scope of the work is, how it affects the involved compounds and what implications it may have for other crystalline network solids.  

It is a valid remark that the scope of this contribution is somewhat narrow for a general chemistry journal. However, this article is contributed to the memorial issue for Howard Flack, who was among other things, known for his contribution to absolute structure determination. We submitted this particular paper with that in mind.

Reviewer 1:

The information below may be useful for future revisions or other journal entries.

1) References have been excluded from many of the sentences in the Introduction. Accordingly, it is essential to add a reference for the reader in the sentence below.

-Line 22: Metal-organic frameworks (MOFs) are porous solids consisting of inorganic nodes linked by organic multidentate ligands (e.g. through strongly coordinating groups such as carboxylates or Lewis bases) to form extended networks.

-Line 24: The combination of diverse coordination chemistry and the structural richness of organic ligands enable a vast number of possible MOF structures and network topologies.

-Line 32: The principal method of MOF structure determination is X-ray diffraction methods, and single crystal X-ray diffraction (SC-XRD) in particular.

-Line 33: Despite being the most powerful technique to solve crystal structures, a successful SC-XRD experiment normally depends on a relatively large single crystal of high quality.

Reply: We regarded the above statements as self-evident. However, we can see that not all of them are for researchers outside of the MOF field. References have been added to the highlighted sentences, except for the last one, which we still regard as self-evident.

Reviewer 1:

2) I suggest a consistent expression in the two sentences below. This will be useful to increase the readability of the reader.

Line 104: …beamline ID11 at the ESRF synchrotron 104 (Grenoble, France),

Line 109: … beamline I911-3 at the MAX2 synchrotron in Lund, Sweden.

Reply: Line 109 was changed for consistency.

Reviewer 1:

3) In Figure 4, the notation (d) overlaps with the figure 4d.

 Reply: The figure has been changed to provide spacing between the character and the figure.

Reviewer 1:

4) Line 187: “… acquired at beamline ID-11 at the European synchrotron (ESRF)” This sentence has already been mentioned in the materials and methods section.

Reply: The sentence has been revised.

Reviewer 1: 5) The reference must appear at the end of the sentence and before the period.

 Reply: This has been corrected throughout the text.

Reviewer 1: 6) Authors should clearly indicate the references below.
Reference 4: The author should indicate the volume and page number. The word "doi" was duplicated.

Reference 6: “doi” was duplicated.

Reference 11-14: “doi” was duplicated.

Reference 19: The author should remove “http://doi.org/”

Reference 22: below sentence should be deleted . “http://www.nature.com/nchem/journal/v8/n3/abs/nchem.2430.html#supplementary-information.”

Reply: We thank the reviewer for spotting these mistakes. They have all been corrected.

Reviewer 1: 7) In order to further emphasize the author's scientific results, it will be important to cover more details about the meaning of this study and how it can be applied to crystal twin in the future in discussion and conclusion.

Reply: We agree that this point was not communicated clearly enough in the original submission. A paragraph has been appended in the introduction to clarify.

Best regards,

Sigurd Øien-Ødegaard

Reviewer 2 Report

The manuscript is recommended for publication after minor revision has been done. The manuscript shows pitfalls which occur frequently in crystal structure determination leading to false structure description. It is highly important to make these problems obvious to a broad readership.

page 2:

"different geometry"  there are no different geometries, what you mean is different molecular shape or similar. The geometry is three-dimensionally euclidean, nothing else.

Table 1:

Do not write F squared. Correct: square of |F|. F is a complex number even in centrosymmetric space groups, if anomalous dispersion is taken into account, which holds true here. The square of a complex number is different from the square of its magnitude.

This is a fact of mathematics and physics even if common computer output does not take care.

That F is a complex number in non-centrosymmetric spacegroups (one is encountered with compound 3) should be common to everyone.

In a manuscript which demonstrates how important correct physical and mathematical treatment of diffraction data is, it is mandatory that everything else is flawless as well.

Author Response

We are pleased to submit the revised manuscript entitled “Twinning in Zr-based Metal-organic Framework Crystals”. We would like to thank the reviewers for providing insightful and constructive criticism which have improved the clarity and quality of the manuscript. Below is a point-by-point summary of the reviewer’s remarks and our replies.

Reviewer 2:

The manuscript is recommended for publication after minor revision has been done. The manuscript shows pitfalls which occur frequently in crystal structure determination leading to false structure description. It is highly important to make these problems obvious to a broad readership.

Reply:  We agree that this point was not communicated clearly enough in the original submission. We have appended a paragraph to the introduction to clarify what the scope of the work is, how it affects the involved compounds and what implications it may have for other crystalline network solids.

There are several examples in the literature of MOF structures that are probably wrong, because twinning has been overlooked in the data reduction. However, we cannot be sure of what mistakes have been made without having seen the raw data. For this reason, we decided not to name any specific publications.

Reviewer 2: page 2:

"different geometry"  there are no different geometries, what you mean is different molecular shape or similar. The geometry is three-dimensionally euclidean, nothing else.

Reply: We thank the reviewer for pointing out this poor choice of words. The sentence was rewritten to clarify what is meant.

Reviewer 2: Table 1:

Do not write F squared. Correct: square of |F|. F is a complex number even in centrosymmetric space groups, if anomalous dispersion is taken into account, which holds true here. The square of a complex number is different from the square of its magnitude.

This is a fact of mathematics and physics even if common computer output does not take care.

That F is a complex number in non-centrosymmetric spacegroups (one is encountered with compound 3) should be common to everyone.

In a manuscript which demonstrates how important correct physical and mathematical treatment of diffraction data is, it is mandatory that everything else is flawless as well.

Reply: We agree with the reviewer on the assessment of F versus |F|. However, Table 1 uses the standard formatting of the Acta Crystallographica journals and its entries must thus be regarded as conventional when reporting crystal structure refinement values (although it is technically wrong). We encourage the editorial office to decide on a policy regarding this issue (and crystallographic table items in general), and will comply with their decision.

Round 2

Reviewer 1 Report

The author responded well to concerns about publication during the revision. Particularly, the lack of meaning of the study was reinforced in the "Introduction section", so the previous reviewer's concerns about the creativity and originality of the study were resolved. Now I consider that the revised manuscript contents may give new insights to other researchers in the future. Accordingly, I suggest the publication of current revised manuscript.